Accepted at the ICLR 2024 Workshop on AI4Differential Equations In Science

# Hessian Reparametrization for Coarse-grained Energy Minimization

**Nima Dehmamy**
IBM Research
Nima.Dehmamy@ibm.com

**Csaba Both**
Northeastern University
both.c@northeastern.edu

**Jeet Mohapatra**
MIT CSAIL
jeetmo@mit.edu

**Subhro Das**
IBM Research
subhro.das@ibm.com

**Tommi Jaakkola**
MIT CSAIL
tommi@csail.mit.edu

## Abstract

Energy minimization problems are highly non-convex problems at the heart of physical sciences. These problems often suffer from slow convergence due to sharply falling potentials, leading to small gradients. To make them tractable, we often resort to coarse-graining (CG), a type of lossy compression. We introduce a new way to perform CG using reparametrization, which can avoid some of the costly steps of traditional CG, such as force-matching and back-mapping. We focus on improving the slow dynamics by using CG to projecting onto slow modes. We show that in many physical systems slow modes can remain robust under dynamics and hence can be pre-computed from the Hessian of random configurations. We show the advantage of our CG method on some difficult synthetic problems inspired by molecular dynamics (MD). We also test our method on molecular dynamics for folding of small proteins, showing modest improvements. We observe that our method either reaches lower energies or runs in shorter time than the baseline non-CG simulations.

In statistical physics we are interested in knowing in the most likely states the system will be in, given a set of exogenous factors (e.g. finding protein conformations at different temperatures, pH levels, etc) (Landau & Lifshitz, 1980). The negative log-likelihood function, or "free energy", is based on physical laws, with more likely states having lower free energies. These systems generally have a large number of degrees of freedom (DOF) with complex interactions, making the free energy landscape high dimensional and highly non-convex. One way to make these problems tractable is to compress the DOF via coarse-graining (CG) (Saunders & Voth, 2013; Noid, 2013). CG aims to reduce the degrees of freedom (DOF) in the system and replace them with clustered or collective modes. CG methods have proven very successful (Pak & Voth, 2018) in many fields such as molecular dynamics (MD) (Hollingsworth & Dror, 2018). Yet, they also involve costly steps, such as "back-mapping", a one-to-many map to fine-grained (FG) modes, and "force-matching" (Jin et al., 2022), i.e. finding the effective forces experienced by CG modes during dynamics.

In this paper we introduce a new approach to CG where instead of FG modes being replaced by CG modes, we reparametrize the FG modes as functions of the CG modes. The idea of this reparametrization is similar to Deep Image Priors (Ulyanov et al., 2018), as the CG modes can become parameters of a neural network whose output is the FG representation of the system. This approach allows us to both have access to FG modes at all times (avoiding back-mapping) and to compute forces without requiring force-matching. We choose our CG modes to be the "slow modes", which are collective modes that evolve very slowly. Slow modes are the main cause of slow convergence of many problems, including MD. While slow modes may change during optimization, we show that in many physical systems, there exist robust slow modes, which change very little during optimization. We use these slow modes for our CG reparametrization. This allows us to first do the hard and slow part of the optimization without interference from fast modes. In the end, we allow all FG modes to relax, including the fast modes, to further optimize the solution. We show experimental results for synthetic force systems with Lennard-Jones potentials and MD simulations,

where we find the reparametrized models achieve deeper energy levels or converge in shorter times, or both. In summary, our contributions are:

1. **CG via reparametrization:** a new method avoiding force-matching and back-mapping.
2. **Robust slow modes:** we devise a method for finding these modes in many systems.
3. **MD simulations:** we show the benefits of our method in protein MD simulations.
4. **Data-free:** our method modifies the optimization process and requires no training data.

Unlike optimization problems encountered in deep neural networks, these physics-based models are "shallow" by construction and usually do not have any input output data. The trainable "weights" are the physical DOF that need to be optimized, and the loss function is the free energy, which encodes all the nonlinear interactions of the weights. Usually, the interactions have multiple scales of strength. For instance, in MD covalent bonds are very strong, modeled as quadratic potentials $\|r - r_0\|^2$, while van der Waals forces between neutral atoms are extremely weak, falling sharply and modeled as Lennard-Jones (LJ) potentials $c(r^{-12} - r^{-6})$. The hierarchy of forces makes the energy minimization quite challenging, including slow convergence of gradient-based methods. To accelerate such free energy minimization problems we need to address the slow convergence of these slow modes. Two commonly taken approaches to address this issue are: 1) preconditioning (e.g. adaptive gradient or quasi-Newton method); 2) Coarse-graining (truncating the DOF to remove some fast modes). We focus on the second option here.

# 1 COARSE-GRAINING BY REPARAMETRIZATION

Let $X \in \mathcal{X} \simeq \mathbb{R}^{n \times d}$ be a set of DOF, e.g. particle positions, bond angles, etc. Let $\mathscr{L} : \mathcal{X} \to \mathbb{R}$ be a loss function, which we will call the energy or potential. Our goal is either to simulate the dynamics of the system based on $\mathscr{L}$, or to find high likelihood configurations $X^*$, which are deep local minima of $\mathscr{L}$. Usually $n$ is large and $\mathscr{L}$ is a very steep non-convex function, making computations slow. Traditional CG involves mapping to a reduced space of CG variables, $\mathcal{Z} \sim \mathbb{R}^{k \times d}$ with $k \ll n$. But to run the dynamics using CG modes we need to figure out forces between CG modes, or "force-matching", and how to go back to $\mathcal{X}$, or "back-mapping".

**Force-matching.** The microscopic energy function is $\mathscr{L} : \mathcal{X} \to \mathbb{R}$. However, as CG replaces the DOF, we need to find an approximate potential $\mathscr{L}_{CG} : \mathcal{Z} \to \mathbb{R}$ such that for $X \in \mathcal{X}$ we have

$$\text{CG:} \quad \phi : \mathcal{X} \to \mathcal{Z}, \quad \mathscr{L}_{CG}(\phi(X)) \approx \mathscr{L}(X). \tag{1}$$

The process of finding $\mathscr{L}_{CG}$ is called force-matching. Traditional force-matching methods involve analytically solving and approximating $\mathscr{L}_{CG}$ Jin et al. (2022). Recently, machine learning (ML) has been used to do force-matching for MD with good results Majewski et al. (2023); Krämer et al. (2023). Next, the dynamics is run using $\mathscr{L}_{CG}$ instead of $\mathscr{L}$.

**Back-mapping.** After the dynamics is run, we need to map back from $\mathcal{Z}$ to $\mathcal{X}$. However, the map $\rho : \mathcal{Z} \to \mathcal{X}$ is not unique, as generally many different $X$ can be compressed to a given CG mode $Z$. Also, when going from $Z$ to a possible $X$, some $X$ may not be allowed (e.g. have overlapping atoms) or have large energies. Therefore, back-mapping usually involves sampling or optimization to find the allowed $X$. This can be challenging when there are many $X$ mapping to the same $Z$. For instance, in protein dynamics most CG methods replace all atoms in each side chains with a single bead at their center of mass.

## 1.1 CG USING REPARAMETRIZATION

Our idea is to change the DOF $X$ to a function of the CG modes, meaning

$$\textbf{Reparametrization:} \quad X = \rho(Z), \quad \rho : \mathcal{Z} \to \mathcal{X} \tag{2}$$

which is the reverse of what traditional CG does. The advantages of this approach are:

1. **No back-mapping:** we have direct access to the fine-grained modes as $X = \rho(Z)$.
2. **No force-matching:** The energy for CG modes is $\mathscr{L}_{CG}(Z) = \mathscr{L}(X) = \mathscr{L}(\rho(Z))$.

$\rho$ may be parametrized as a neural network. In this case, $Z$ represents the parameters of the neural network. In the simplest case, these parameters may just be linear coefficients of the CG modes (see below). But, in general, $\rho$ may be a deep neural network (DNN), similar to the approach taken in Deep Image Priors (Ulyanov et al., 2018). We want this DNN to use the CG modes in a central way to improve upon the original FG optimization. We discuss below how this can be achieved by defining a special graph neural network.

One disadvantage of this reparametrization approach is that we still use $X$ to compute $\mathscr{L}_{CG}(Z)$. Thus, unlike traditional CG, CG reparametrization makes every step of the CG optimization more expensive than FG. Nevertheless, if the CG leads to fewer optimization steps overall, or allows us to find deeper minima more efficiently, the benefits can outweigh the costs. Additionally, this cost could be reduced if $\mathscr{L}_{CG}(Z)$ could be learned from $\mathscr{L}(X)$. We will not do so in this work, but discuss this as a future direction. Next we discuss the reparametrization function $\rho$.

**Choice of reparametrization.** As mentioned above and elaborated in sec. D, a major bottleneck is energy minimization are the "slow modes". If we ignore fast modes in early stages of the optimization, we could traverse much larger time intervals. Therefore, we choose our reparametrization $\rho$ to be simply a linear projection onto slow modes.
**Slow mode projection:**

$$X = \rho(Z) = Z^T \Psi_{\mathbf{Slow}} \equiv \sum_{i \in \mathbf{Slow}} Z_i^T \psi_i \qquad (3)$$

where $\Psi = (\psi_i)_{i \in \mathbf{Slow}}$ form a basis for **Slow**, the subspace spanned by slow modes.

**Graph Neural Networks (GNN):** In a recent work Both et al. (2023) showed that in optimization problems over graphs, using a GNN to reparametrize the node states can lead to significant improvements, reaching both lower losses as well as faster convergence. Inspired by this, we also experimented with GNN reparametrization $X = \text{GNN}(Z)$. The idea is that the Hessian "backbone", described below, can serve as a weighted graph adjacency matrix and used for message passing. In this case, $Z$ consists of the GCN weights, the MLP weights, and an $n \times h_0$ matrix of random $h_0$ dimensional features for each node which are also optimized. The details of our GNN architecture are described in the sec. 2.

## 1.2 The Hessian backbone and robust slow modes

As we show in sec. E, the slow modes are Hessian eigenvectors with eigenvalues close to zero. But since the Hessian $H(X) = \boldsymbol{\nabla}\boldsymbol{\nabla}\mathscr{L}(X)$ depends on $X$, the slow modes can change during the dynamics. Nevertheless, we show that many physical energies, including in MD, have a special structure which allows us to find a set of robust slow modes (app. E). These slow modes should remain reliable for a long interval during optimization.

**Aggregating sampled Hessians.** To find a set of robust, approximate slow modes, we first compute the Hessian for a few perturbed configurations $\mathbf{Samples}(X) = \{X' = X + \delta X\}$, where $X$ is the current state and $\delta X_i^\mu \sim \mathcal{N}(0, \sigma)$ is sampled from Gaussian ($i$ is the particle index and $\mu$ the spatial dimension). We want to extract a set of slow modes from the sampled Hessians $H(X')$. We then compute a backbone from these Hessians of the form

$$\text{Backbone:} \quad \mathbf{H}_{ij} = \sum_{X' \in \mathbf{Sample}(X)} \|H_{ij}(X')\|^2 \qquad (4)$$

Here $i, j \in \mathbb{Z}_n$ are the particle indices and the Frobenius norm $\|H_{ij}\|^2 = \sum_{\mu,\nu}(H_{ij}^{\mu\nu})^2$ sums over the feature indices (note that $X_i^\mu$ has a particle index $i$ and a feature index $\mu \in \{1, \ldots d\}$). Then, we extract the slow modes of the backbone, by doing a spectral expansion $\mathbf{H} = \sum_i \lambda_i \psi_i \psi_i^T$ and picking $\psi_i$ with $|\lambda_i| < \varepsilon^2 \max_j[\lambda_j]$, for some small $\varepsilon < 1$. The intuition behind equation 4 is to identify the components in the sampled Hessians which have consistently high magnitudes.

## 2 Experiments

We apply our method to protein folding using classical MD forces.
**Settings:** We use gradient descent to minimize $\mathscr{L}(X)$. All experiments (both CG and baseline) use

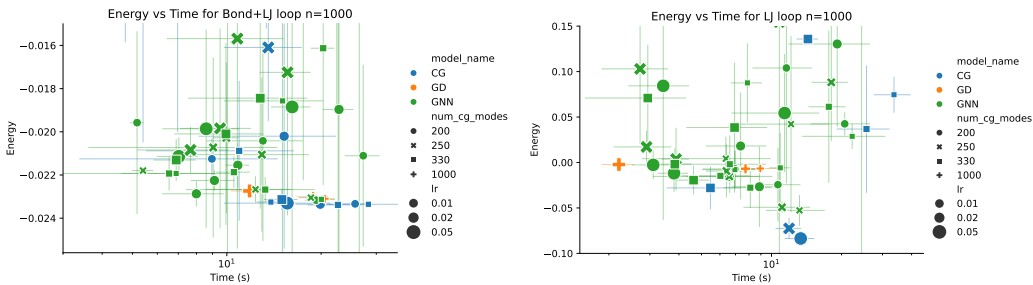

Figure 1: **Synthetic loop folding** ($n = 1000$). In Bond+LJ (left), a quadratic potential $\sum_i (r_{ii+1} - 1)^2$ attracts nodes $i$ and $i + 1$. A weak LJ potential attracts nodes $i$ and $i + 10$ to form loops. In LJ loop (right) both the backbone $i, i + 1$ and the 10x weaker loop are LJ. Orange crosses denote the baseline GD, green is GNN and blue is CG. The dots are different hyperparameter settings (LR, Nr. CG modes, stopping criteria, etc.) with error bars over 5 runs. In Bond+LJ, CG yields slightly better energies but takes longer, while GNN can converge faster to GD energies. In pure LJ, using CG and GNN can yield significantly better energies.

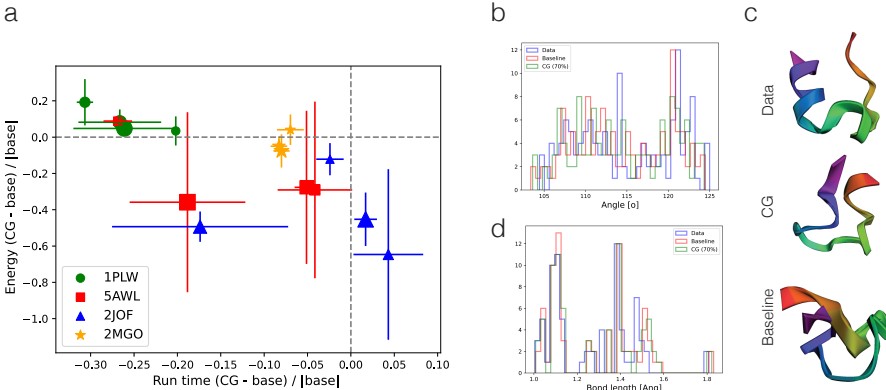

Figure 2: (a) time vs. energy of CG runs relative to the baseline. The symbol sizes encode different numbers of eigenvectors in CG. The different colors and symbols correspond to different proteins: 5AWL, 1PLW, 2JOF, and 2MGO. (b) The angle and (d) bond length distribution of 1PLW protein acquired from the data and calculated by using the baseline and CG (70%) methods. (c) The native structure (data) of the 2JOF protein, and the layouts obtained from the baseline and CG method.

the Adam optimizer with a learning rate $10^{-2}$ and early stopping with $|\delta\mathscr{L}| = 10^{-6}$ tolerance and 5 steps patience. We run each experiment four times.

**Baseline:** we use gradient descent (GD) with Adam optimizer on the MD energy as baseline.

**CG model:** We use four different choices for the fraction of the eigenvectors to use in CG equation 3: $3 \times (\#\text{AminoAcids})$, 30%, 50%, and 70%. We use a two stage process. First, we use CG as in equation 3 $X = \rho(Z) = Z^T \Psi_{\textbf{Slow}}$ and minimize $\mathscr{L}_{CG}(Z) = \mathscr{L}(\rho(Z))$ over $Z$. After convergence to $X_0 = \rho(Z_0)$, we add $\delta X$ to $X_0$ and optimize the fine-grained $\delta X$, starting with $\delta X = 0$.

**GNN model:** We use a GNN consisting of a graph convolution (GCN) layer with self-loops and one node-wise MLP layer, projecting the GNN output to 3D to get particle positions. The GCN takes $Z_{h_0} \in \mathbb{R}^{n \times h_0}$ as input, with $h_0 > 3$ and has weights $W_G \in \mathbb{R}^{h_0 \times h_1}$. Then, GCN output gets a Tanh activation and is passed to the MLP layer to yield $X$. The CG parameters in this case are $Z_h, W_G$ and the weights and biases of the MLP.

**Synthetic coil:** We use quadratic and LJ potentials to make synthetic systems whose minimum energy state should be a coil (looping every 10 nodes), inspired by MD potentials. Figure 1 shows many experiments using GD, CG and GNN. In the quadratic Bond+LJ case GNN yields good speedup, while CG yields better energies. The benfits of CG and GNN become more apparent in the harder pure LJ problem, where GD fails to find good energies, while CG finds very much deeper energies, followed by GNN (Fig. 3).

**Protein folding with classical MD:** We implement a simplified force-field with implicit solvent (i.e. water molecules are not modeled and appear as hydrogen-bonding and hydrophobicity terms; app. B). In protein folding our energy function consists of five potential energies for: bond length $E_{bond}$, bond angles $E_{angle}$, Van der Waals $E_{vdW}$, hydrophobic $E_{hp}$ and hydrogen bonding $E_H$ Ceci

et al. (2007). Figure 4 shows an example of these coupling matrices for the Enkephalin (1PLW) protein. To evaluate the effect of our CG model, we run experiments on four small proteins: Chignolin (5AWL), Trp-Cage (2JOF), Cyclotide (2MGO) and Enkephalin (1PLW).

**Protein results:** Denoting the final energy and run time of the CG model by $E_{CG}$ and $t_{CG}$, and baseline by $E_0$ and $t_0$, we compute the normalized differences $\delta\hat{E} = (E_{CG} - E_0)/E_0$ and $\delta\hat{t} = (t_{CG} - t_0)/t_0$, to plot different proteins together. Figure 2a shows the mean of $\delta\hat{E}$ vs $\delta\hat{t}$ over the 4 runs for each CG setting (errorbars are 1 STD). Overall we find that all CG models outperform the baseline either in terms of run time or energy, or both.

## DISCUSSION

We Showed preliminary evidence that CG through reparametrization can yield some improvements over non-CG baseline in protein folding, both in terms of run time as well as energy. This method has the advantage that it does not require force-matching or back-mapping. However, more experiments are needed to compare it against traditional CG methods. In fact, using ML to learn force-matching might provide further advantage by removing the need to evaluate $\mathscr{L}_{CG}(Z) = \mathscr{L}(X)$ via the fine-grained modes $X$. Also, while our canonical slow modes are derived for physical Hessians, the reparametrization approach to CG is general and could be applied to other ML problems.

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

# A  FINDING EFFECTIVE DOF

We want to see if the Hessian can help us find effective dof like the $\phi, \psi$ angles in alanine dipeptide. The core idea is to look for symmetries of the Hessian CG modes. Assume we are close to a minimum $x_*$ in the energy landscape. Using a Taylor expansion and the fact that $\boldsymbol{\nabla} E(x_*) = 0$, the energy can be approximately be written as

$$E(x_* + x) \approx E(x_*) + \mathrm{Tr}\{x^T \boldsymbol{H} x\} + O(x^3) \tag{5}$$

Here, $x$ are usually expected to be small vibrational modes. However, if the system has symmetries, $x$ can be quite large and still keep the system in the basin of the same minimum. For example, when the energy is translation invariant, $x$ can be translated arbitrarily to $x' = x + \delta x$ without changing the energy. We are interested in finding such large transformations that go beyond small oscillations. In MD, the energies are generally invariant under a global $SE(3)$. But the energy near a minimum may have other, "local" invariances. For instance, in alanine dipeptide, the two peptides can be rotated around the backbone without incurring a large energy cost. The rotations can also be arbitrarily large. This type of large modes is what we are concerned with here.

We want to see if there are ways that the conformation can change without costing much energy. We can also frame this as a question of symmetries of the Hessian $\boldsymbol{H}$. Suppose that we construct $x$ using a small linear transformation $\boldsymbol{U} = I + \varepsilon \boldsymbol{L}$ on $x$, meaning

$$x' = \boldsymbol{U} x = x + \varepsilon \boldsymbol{L} x = x + \delta x \tag{6}$$

where $\varepsilon \ll 1$ is a small number and $\boldsymbol{L}$ is a linear transformation of unit Frobenius norm. The condition that $x$ does not change the energy yields

$$
\begin{aligned}
E(x_* + x) &\approx E(x_* + x') \\
\Rightarrow \mathrm{Tr}\{x^T \boldsymbol{H} x\} &\approx \mathrm{Tr}\{x^T \boldsymbol{U}^T \boldsymbol{H} \boldsymbol{U} x\} \\
&= \mathrm{Tr}\{x^T \boldsymbol{H} x\} + \varepsilon \, \mathrm{Tr}\{x^T \left(\boldsymbol{L}^T \boldsymbol{H} + \boldsymbol{H} \boldsymbol{L}\right) x\} + O(\varepsilon^2)
\end{aligned}
\tag{7}
$$

To satisfy this relation for an arbitrary vibration $x$, we need to find $L$ such that

$$x^T \left(\boldsymbol{L}^T \boldsymbol{H} + \boldsymbol{H} \boldsymbol{L}\right) x = x^T \left(\boldsymbol{H}^T + \boldsymbol{H}\right) \boldsymbol{L} x = O(\varepsilon) \tag{8}$$

Any $\boldsymbol{L}$ that satisfies equation 8 for arbitrary $\boldsymbol{x}$ is a symmetry of the Hessian. This is equivalent to restricting the operator norm of $(\boldsymbol{H}^T + \boldsymbol{H})\boldsymbol{L}$ to be $O(\epsilon)$. We establish the following result, that shows that the slow modes of the Hessian form the most dominant modes of the symmetry.

**Theorem A.1.** *For a given symmetric matrix $\boldsymbol{A}$ with $\max_{\|x\|=1} \|\boldsymbol{A}x\| \leq 1$, with eigendecomposition given as*

$$\boldsymbol{A} = \boldsymbol{V}\Lambda\boldsymbol{V}^T = \boldsymbol{V}_\epsilon \Lambda_\epsilon \boldsymbol{V}_\epsilon^T + \boldsymbol{V}_1 \Lambda_1 \boldsymbol{V}_1^T$$

*with slow modes given by $\boldsymbol{V}_\epsilon$ with corresponding eigenvalues $\Lambda_\epsilon, \left(|\lambda_i^\epsilon| = O(\epsilon)\right)$ and fast modes given as $\boldsymbol{V}_1$ with eigenvalues $\Lambda_1, \left(|\lambda_i^1| = \Omega(1)\right)$ we see that for any unit Frobenius norm approximate symmetry generator matrix $\boldsymbol{B}$ satisfying,*

$$\|\boldsymbol{A}\boldsymbol{B}\|_{op} \leq \min_{\|\boldsymbol{M}\|_F = 1} \|\boldsymbol{A}\boldsymbol{M}\|_{op} + O(\epsilon)$$

*we have that for all vectors $x$*

$$\left\|\boldsymbol{B}\left(\boldsymbol{V}_1 \boldsymbol{V}_1^T x\right)\right\|_2 \leq O(\epsilon)\|x\|_2 \tag{9}$$

Considering a given value of $\boldsymbol{H} + \boldsymbol{H}^T$, there are many linear transformations $\boldsymbol{L}$ that satisfy equation equation 8 . We restrict our attention to only transformations of the form :

$$\boldsymbol{L}_P = \sum_{i=1}^{N} \lambda_i \cdot v_i v_i^T \otimes \boldsymbol{L}_i \quad \text{with} \sum_i \lambda_i^2 = 1 \tag{10}$$

where $\boldsymbol{L}_i$'s are distance preserving transformation matrices in the spatial dimensions. In this formulation, the $v_i$ vectors give directions of partial symmetry for the $N$-particle system and the condition

over $\lambda_i$ ensures that the Frobenius norm of $\boldsymbol{L}_P$ is 1. Using equation 9 we see that it suffices to only restrict our attention to the slow modes of the Hessian, $V_\epsilon$.

$$x^T \left(L_P^T \boldsymbol{H} + \boldsymbol{H} L_P\right) x = O(\varepsilon)$$

$$
\begin{aligned}
x^T \left(L_P^T \boldsymbol{H} + \boldsymbol{H} L_P\right) x &= \left(\sum_{i=1}^N \lambda_i c_i \cdot v_i \otimes (L_i r_i^x)\right)^T \left(\boldsymbol{H}^T + \boldsymbol{H}\right) \left(\sum_{i=1}^N c_i \cdot v_i \otimes r_i^x\right) \\
&= \sum_{i,j} \lambda_i c_i c_j \cdot (v_i \otimes L_i r_i^x)^T \left(\boldsymbol{H}^T + \boldsymbol{H}\right) \left(v_j \otimes r_j^x\right) \\
&= \sum_{i,j} \lambda_i c_i c_j \cdot (L_i r_i^x)^T \left(v_i^T \boldsymbol{H}^T v_j + v_i^T \boldsymbol{H} v_j\right) r_j^x \\
&\leq \sqrt{\sum_{i,j} \lambda_i c_i c_j \cdot \left\| v_i^T \left(\boldsymbol{H}^T + \boldsymbol{H}\right) v_j \right\|_F^2}
\end{aligned}
\tag{11}
$$

where for the last inequality we use the fact that the Frobenius norm is larger than the operator norm and $\boldsymbol{L}_i$ are distance preserving matrices which also preserves the Frobenius norm.

## B  PROTEIN FOLDING WITH CLASSICAL MD

In protein folding our energy function consists of five potential energies for: bond length $E_{bond}$, bond angles $E_{angle}$, Van der Waals $E_{vdW}$, hydrophobic $E_{hp}$ and hydrogen bonding $E_H$ Ceci et al. (2007). Note that we are ignoring the solvent (e.g. water) and writing using potentials, or force fields. To calculate the force field, we use distance, $r$, and angle-based, $\Theta$, potentials. For each amino acid, we use the rdkit Landrum et al. (2020) package to acquire bond length, $r_0$, and bond angle, $\theta_0$ (every triplet of atoms defining the bond), information that we use to define quadratic energies $E_{bond}$ and $E_{angle}$. We use Lennard-Jones (LJ) potentials, $V_{p,q}(r) = r^{-p} - r^{-q}$, to approximate $E_{vdW}$ between all pairs of atoms, $E_H$ between atoms prone to form a hydrogen bond (certain $H$ and $O$, in our case), $E_{hp}$ between atoms in hydrophobic residues, yielding

$$
\begin{aligned}
\mathscr{L}(X) =& E_{bond} + E_{angle} + E_{vdW} + E_H + E_{hp} \\
=& k_{bond}(r - r_0)^2 + k_{angle}(\theta - \theta_0)^2 \\
& + \epsilon_{vdW} V_{12,6}\left(\frac{r}{\sigma_{vdW}}\right) + \epsilon_H V_{6,4}\left(\frac{r}{\sigma_H}\right) + \epsilon_{hp} V_{6,4}\left(\frac{r}{\sigma_{hp}}\right)
\end{aligned}
\tag{12}
$$

Here the coupling matrix $[\sigma_{vdW}]_{ij} = a_i + a_j$ where $a_i$ is the vdW radius of atom $i$. For atoms which form H-bonds, $[\sigma_H]_{ij} = (b_i \cdot b_j)1.5\text{Å}$ (hydrogen bonding radius) with $b_i = 1$ if $i$ forms an H-bond, and $b_i = 0$ otherwise. $[\sigma_{hp}]_{ij} = c_i + c_j$ where $c_i = 2\text{Å}$ if atom $i$ is in a hydrophobic residue and $c_i = 0$ otherwise.

We note that our choices for $\epsilon_H, \epsilon_{vdW}, \epsilon_{hp}$ and $k_{bond}, k_{angle}$, can be a source of error. Additionally, we "softened" the LJ potential to $V_{p,q} = 1/(r^p + \zeta) - 1/(r^q + \zeta)$ with $\zeta = 0.65$, which is large and significantly reduces the penalty for overlapping atoms and may reduce accuracy.

## C  ADDITIONAL FIGURES

## D  ENERGY MINIMIZATION

Let $X \in \mathcal{X} \simeq \mathbb{R}^{n \times d}$ be a set of degrees of freedom (e.g. particle positions, bond angles, etc.) and let $\mathscr{L} : \mathcal{X} \to \mathbb{R}$ be the energy (loss) function. We are interested in finding configurations $X^*$ which are local minima of $\mathscr{L}$. We can find such $X^*$ using a gradient descent (GD), or its continuous variant, gradient flow (GF)

$$\frac{dX}{dt} = -\varepsilon \boldsymbol{\nabla} \mathscr{L}(X) \tag{13}$$

where $\varepsilon$ is the matrix of learning rates (LR). In simple GD where $\varepsilon = cI$ is a single constant times identity, GD evolves at different rates in different directions, with some being much slower than

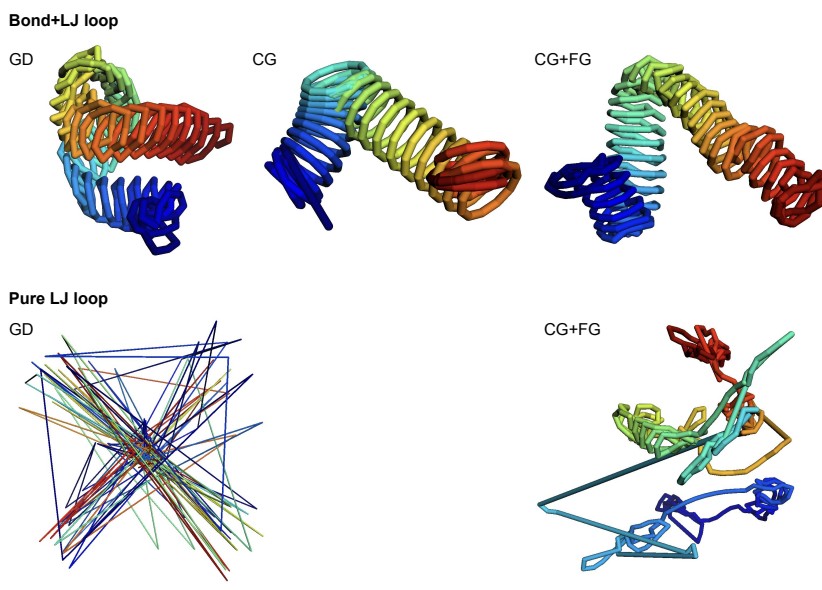

Figure 3: Example runs of the synthetic loop experiments with $n = 400$ nodes. Top row is the Bond+LJ and bottom the pure LJ experiments. In Bond+LJ, GD already finds good energies and the configuration is reasonably close to a loop, though flattened. When using CG, the initial CG stage yields very good coil fragments, but has some overlaps. The final fine-graining (CG+FG) resolves these crossings. The pure LJ case is much more tricky. But in most runs, GD completely fails to get close to a deep energy minimum and is stuck at very high energy states, with no sign of the loop forming. Using CG+FG, however, yields much better results, with small loop fragments forming.

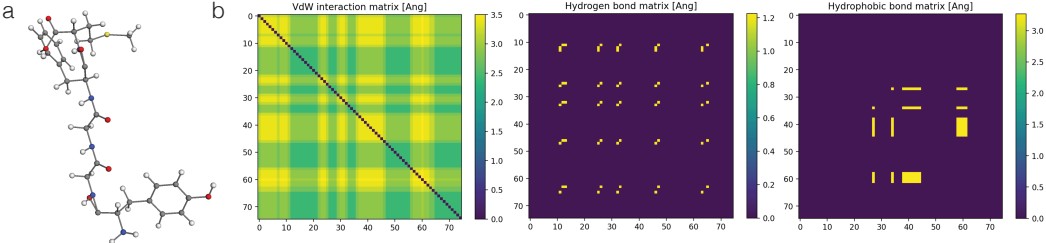

Figure 4: Enkephalin (1PLW). a) The peptide chain is built by stacking amino acids on each other using the peptide bond length from the literature, $1.32\mathring{A}$. b) Van der Waals, hydrogen bond, and hydrophobic interaction matrix, that we use in the energy optimization.

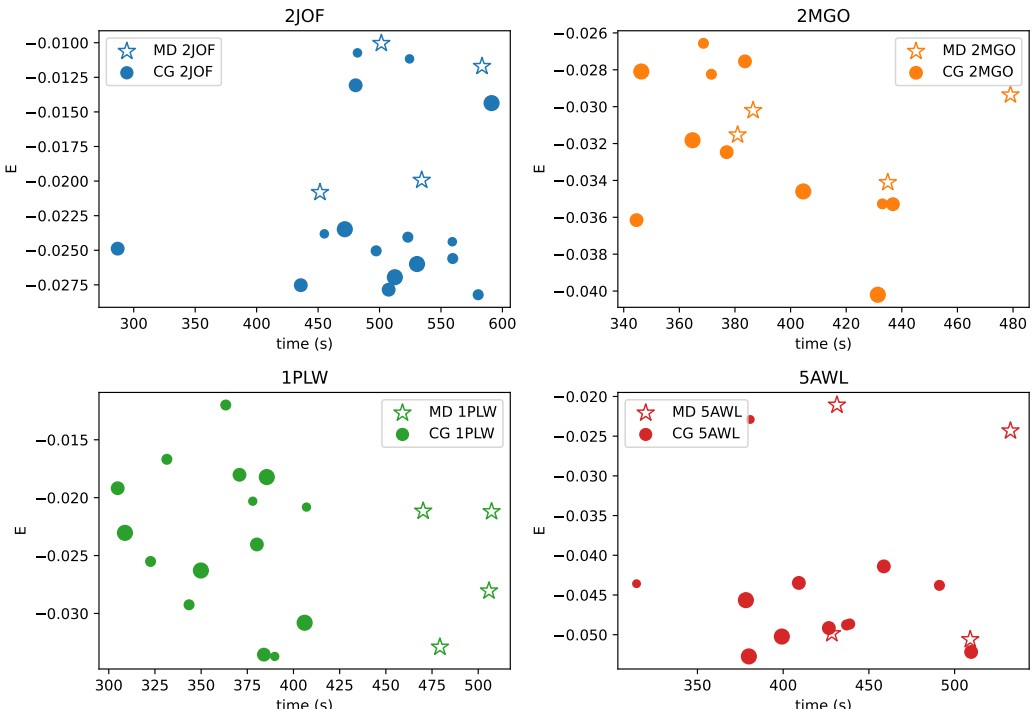

Figure 5: Comparison of performance of CG Hessian versus baseline MD. Point sizes correspond to the number of CG modes used.

others. At a given $X$, these "slow modes" are the eigenvectors of the Hessian $H(X) = \boldsymbol{\nabla}\boldsymbol{\nabla}\mathscr{L}(X)$ with eigenvalues closest to zero, as we review below. We will first define fast and slow modes in the simple quadratic case and then generalize them to non-convex cases in the next section.

**Fast and slow modes for quadratic Loss.** Consider the case where $\mathscr{L}(X) = \frac{1}{2}\operatorname{Tr}\{X^T \boldsymbol{H} X\}$. Here $\boldsymbol{H}$ is a Hermitian matrix and the Hessian of $\mathscr{L}$, with a spectral expansion given by $\boldsymbol{H} = \sum_i \lambda_i \psi_i \psi_i^T$, $\lambda_i \in \mathbb{R}$ and $\psi_i \in \mathbb{R}^n$. In this basis we have $X(t) = \sum_i c_i(t)\psi_i$ with $c_i : \mathbb{R} \to \mathbb{R}^d$. Projecting equation 13 onto one of the eigenmodes we get

$$\frac{dc_i}{dt} = \psi^T \frac{dX}{dt} = -\varepsilon\lambda_i \psi^T X = -\varepsilon\lambda_i c_i \tag{14}$$

where we assumed $d\psi_i/dt = 0$. From equation 14 we see that the decay/growth rate along mode $\psi_i$ is $|\varepsilon\lambda_i|$. Hence, modes with $\lambda_i$ close to zero are the "slow modes", evolving very slowly, and large $|\lambda_i|$ defines the "fast modes". Since $c_i(t) = c_i(0)\exp[-t/\tau_i]$ with time scale $\tau_i = 1/(\varepsilon\lambda_i)$, the fast modes evolve exponentially faster than slow modes. This disparity in the rates results in slow

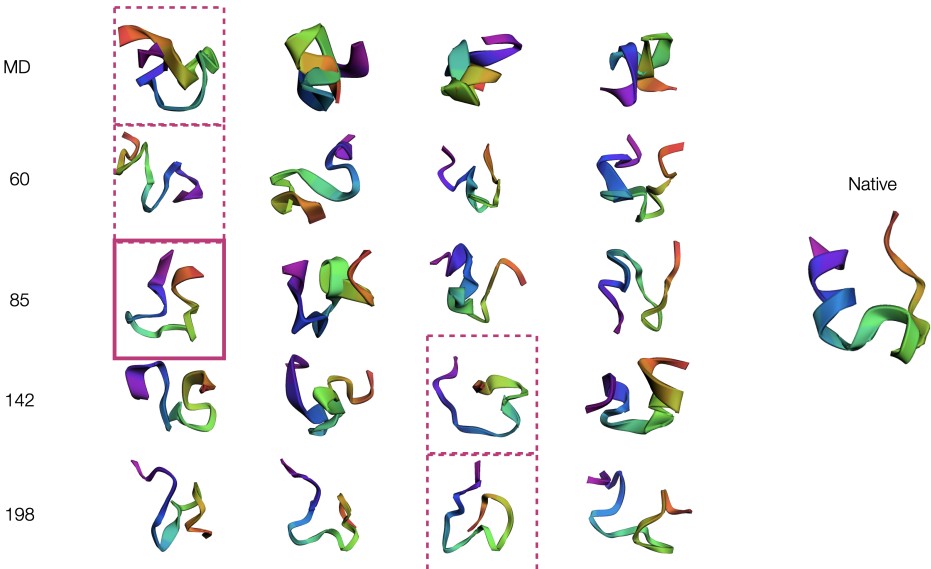

Figure 6: The folded structures of the 2JOF protein by using the CG and baseline method. The numbers in front of the rows are the numbers of eigenvectors used in the CG reparametrization. Dashed frames show the minimum energy embedding in each case, while the thick line frame highlights the absolute minimum layout.

convergence, because the fast modes force us to choose smaller $\varepsilon$ to avoid numerical instabilities. Two potential ways to fix the issue with disparity in time scales are: 1) make rates isotropic (second-order methods and adaptive gradients); 2) mode truncation or compression (CG). We will briefly review the former here.

**Adaptive gradient and second-order methods.** Newton's method uses $\varepsilon = \eta H(X)^{-1}$ which makes GD isotropic along all modes, but it is expensive ($O((3n)^3)$) in our case). Quasi-Newton methods, e.g. BFGS Fletcher (2013), approximate $H^{-1}$ iteratively, but are generally also slow. Another, more efficient approach is adaptive gradient methods, such as AdaGrad Duchi et al. (2011) and Adam Kingma & Ba (2014) which approximate $H$ by $\sqrt{g_t g_t^T + \eta}$ where $g_t = \sum_{i=1}^{k} \gamma^i \boldsymbol{\nabla}\mathscr{L}(X(t - i))$ is some discounted average over past gradients and $\eta$ a small constant. For efficiency, in practice we only use the diagonal part of this matrix to approximate $H^{-1}$. As we will see in experiments, this approximation, while being far superior to GD with constant LR, is still very slow for MD tasks.

Most second-order methods are designed to work for generic problem and don't make strong assumptions about the spectrum of the Hessian. Recent second-order methods such as K-FAC Martens & Grosse (2015) and Shampoo Gupta et al. (2018) work with block diagonal approximations of the Hessian (or the Fisher information matrix), which usually emerges in deep learning models due to model architecture. Instead, we will exploit the spectral properties of the Hessian in physics problems. Fast and slow modes generally arise in physics due to vastly different strengths in forces (e.g. weak van der Waals vs strong chemical bonds).

## D.1 GENERALIZED FAST AND SLOW MODES

The notion of fast and slow modes is helpful for the analysis of any time slice of the dynamics during which the Hessian is not changing dramatically. Consider a configuration $X(t)$ and let $\delta t$ be a small time interval. We are looking for modes which are almost stationary over $\delta t$. To identify these modes, we can for instance find perturbations $\delta X$ which would have almost zero dynamics. concretely we find the dynamics of $X + \delta X$ as

$$\frac{d}{dt}(X + \delta X) = -\varepsilon \boldsymbol{\nabla}\mathscr{L}(X + \delta X) \approx -\varepsilon \boldsymbol{\nabla}\mathscr{L}(X) - \varepsilon H \delta X + O(\delta X)^2 \tag{15}$$

meaning, a small $\delta X$ adds $\varepsilon H \delta X$ to the dynamics.

Thus if $\delta X$ is a zero mode of the Hessian, $H\delta X = 0$, it won't change the dynamics of $X$. To define slow modes, we can slightly relax this and look for normalized modes $\psi = \delta X/\|\delta X\|$ whose associated time scale is much longer than a desired time scale $\delta t$

$$\tau = |\varepsilon\psi^T H\psi| = |\varepsilon\lambda| \gg \delta t \tag{16}$$

which just means that we need to find the approximate zero modes of the Hessian $H(X)$.

**CG by projecting to the slow manifold.** Because the dynamics of the modes above is very slow over $\delta t$, we can safely increase the time scale and run their dynamics over much longer periods $\Delta t \gg \delta t$. The essence of our algorithm is to ignore fast modes and project and evolve the system on the "slow manifold" spanned by the slow modes of the Hessian. However, the main challenge is how to deal with the fact that the Hessian is not constant and depends on the configuration $X$. We address this point next. We show that for a large class of physical potentials one can find a reliable set of approximate slow modes.

## E  PROPERTIES OF PHYSICAL HESSIANS

**Invariant potentials.** In systems of interacting particles in physics, most of the leading interactions are pairwise and involve relative features, $\boldsymbol{r}_{ij} \equiv X_i - X_j$ (distance vector, relative angle, etc). Moreover, they are often invariant under certain global symmetries, such as Euclidean symmetries (translation and rotation) or Lorentz symmetry (relativistic particles). These symmetries keep some 2-norm of vectors, $v^2 = \|\boldsymbol{v}\|_\eta \equiv \boldsymbol{v}^T\eta\boldsymbol{v}$ invariant. Here $\eta$ may be the Euclidean metric $\eta = \mathrm{diag}(1,1,1)$ or the Minkowski metric $\eta = \mathrm{diag}(-1,1,1,1)$ for relativistic problems, etc. For example, the Euclidean norm $\boldsymbol{v}^T\boldsymbol{v}$ in $d$ dimensions is invariant under rotations $\boldsymbol{v} \to g\boldsymbol{v}$, where $g \in SO(d)$, and the Minkowski norm is invariant under the Lorentz group $SO(1, d-1)$.

Let $r$ denote the matrix of distances with $r_{ij} = \|\boldsymbol{r}_{ij}\|_\eta$. Any function of $r_{ij}$ is invariant under symmetries that keep $\|\cdot\|_\eta$ invariant. A general invariant energy function can combine $r_{ij}$ for different $i, j$ in arbitrary ways. Usually in physical systems each pair contributes an additive term in to the total energy. Assuming additivity, the energy has a form

$$\mathscr{L}(X) = \sum_{ij} f_{ij}(r_{ij}) \tag{17}$$

where $f_{ij}(z) = f_{ji}(z)$ (symmetric under $i \leftrightarrow j$). For example, when particle $i$ has electric charge $q_i$, the Coulomb potential between $i, j$ can be written as in equation 17 using $f_{ij}(z) = kq_iq_j/z$. Similarly, weak van der Waals (vdW) forces in molecular systems, which are modeled as Lennard-Jones potential, are also of the form in equation 17 with

$$\text{van der Waals:} \quad f_{ij}(r_{ij}) = V_{p,q}\left(\frac{r_{ij}}{\sigma_{ij}}\right), \quad V_{p,q}(r) = \frac{1}{r^p} - \frac{1}{r^q}. \tag{18}$$

Here $\sigma_{ij} = a_i + a_j$, where $a_i$ is the vdW radius of particle $i$, and vdW uses $p = 12, q = 6$. Next, we show that the Hessian of equation 17 has an important property which aids in finding its slow modes.

### E.0.1  HESSIAN OF INVARIANT POTENTIALS

The Hessian of potentials of the form equation 17 has the special property that it is the graph Laplacian of a weighted graph which depends on $X$, as we show now (see appendix F for details). This will play a crucial role in our argument about canonical slow modes.

**Hessian as a graph Laplacian.** Let $\partial_i \equiv \partial/\partial X_i$ and let $\hat{r} = \eta\boldsymbol{r}/r$ be the dual unit vector of $\boldsymbol{r}$. First, observe that $\partial_i r_{jk} = \hat{r}_{jk}(\delta_{ij} - \delta_{ik})$ where $\hat{r}_{jk}$ is the unit vector of $\boldsymbol{r}_{jk}$ and $\delta_{ij}$ is the Kronecker delta (1 if $i = j$, 0 otherwise). Let $\mathrm{Hes}[g]$ denote the Hessian of a function $g$. We find that (app. F)

$$\mathrm{Hes}[\mathscr{L}](X)_{ij} = \partial_i\partial_j\mathscr{L}(X) = \sum_k (\delta_{ij} - \delta_{jk})\boldsymbol{H}_{ik}(X) \tag{19}$$

where $\boldsymbol{H}_{ik}(X) = \text{Hes}[f_{ik}](r_{ik})$ and given by

$$\boldsymbol{H}_{ik}(X) = \left[\left(f''_{ik}(v) - \frac{f'_{ik}(v)}{v}\right)\hat{v} \otimes \hat{v} + \frac{f'_{ik}(v)}{v}\eta\right]_{v=r_{ik}} \tag{20}$$

Note that $\boldsymbol{H}$ has four indices, with components $\boldsymbol{H}_{ij}^{\mu\nu}$, having two particle indices $i, j$ and two spatial indices $\mu, \nu$. Recall the Laplacian of an undirected graph with adjacency matrix $A$ is defined as $L = \text{Lap}(A) = D - A$, where $D$ is the degree matrix with elements $D_{ij} = \delta_{ij}\sum_k A_{ik}$. The components of Laplacian can also be written as $L_{ij} = \sum_k A_{ik}(\delta_{ij} - \delta_{jk})$. Thus, we see that the Hessian of $\mathscr{L}$ is indeed the Laplacian of $\boldsymbol{H}$

$$\text{Hes}[\mathscr{L}](X)_{ij} = \sum_k (\delta_{ij} - \delta_{jk})\boldsymbol{H}_{ik} = \text{Lap}(\boldsymbol{H})_{ij} \tag{21}$$

where for every pair of spatial indices the Hessian is a Laplacian over particle indices. The Hessian being Laplacian has an important effect on its null eigenvectors. To show this we make use of the incidence matrix.

### E.1   CANONICAL BACKBONE FOR THE HESSIAN

As the Hessian depends on $X$, it is not clear whether slow modes found at a given $X$ would be applicable to other $X$. We need some guarantee that a set of modes exist which are approximately slow modes for the Hessian at a range of different $X$. We could use multiple perturbed configurations $X + \delta X$ with random $\delta X \sim \mathcal{N}(0, T)$ to get an ensemble of Hessians $\mathcal{H} = \{H(X + \delta X)\}$ and find the overlap of the slow modes of the Hessians in $\mathcal{H}$. However, this is expensive, roughly $O(mkn^2)$ for $m = |\mathcal{H}|$ and $k$ slow modes. We cannot recompute the Hessian slow modes often. We also want a method which is more efficient than quasi-Newton methods such as BFGS. Our solution is to find a backbone for the sampled Hessians whose slow modes are guaranteed to be approximate slow modes of the actual Hessians. The key observation is that the Hessian in equation 21 is a Laplacian of a weighted graph. We show that the slow modes of weighted Laplacians overlap significantly with their unweighted counterparts.

We want to extract a set of slow modes from the sampled Hessians $H(X')$. We then compute a backbone from these Hessians of the form

$$\text{Backbone:} \quad \mathbf{H}_{ij} = \sum_{X' \in \mathbf{Sample}(X)} \|H_{ij}(X')\|^2 \tag{22}$$

Here $i, j \in \mathbb{Z}_n$ are the particle indices and the Frobenius norm $\|H_{ij}\|^2 = \sum_{\mu,\nu}(H_{ij}^{\mu\nu})^2$ sums over the feature indices (note that $X_i^\mu$ has a particle index $i$ and a feature index $\mu \in \{1, \ldots d\}$). Then, we extract the slow modes of the backbone, by doing a spectral expansion $\mathbf{H} = \sum_i \lambda_i \psi_i \psi_i^T$ and picking $\psi_i$ with $|\lambda_i| < \varepsilon^2 \max_j[\lambda_j]$, for some small $\varepsilon < 1$.

The intuition behind equation 22 is to identify the components in the sampled Hessians which have consistently high magnitudes. If we had taken a simple mean we could get very small values, because the components can fluctuate randomly. Also, if we had taken the variance instead of the norm, we would get zero for quadratic $\mathscr{L}$, where $H$ is constant and has no variance. However, these intuitions do not show that there would be any connection between the modes of the backbone $\mathbf{H}$ and the actual Hessians $H(X')$. Importantly, entries in $H(X')$ have signs, which affects the spectrum, whereas all entries in $\mathbf{H}$ are positive. So why should the spectra of $H$ and $\mathbf{H}$ be related? This is where the structure of $\mathscr{L}$ comes into play. Indeed, as we show below, for many physical $\mathscr{L}$, the slow modes of the backbone $\mathbf{H}$ approximate the slow modes of sampled $H(X')$ up to $O(\varepsilon^2)$ errors.

**Definition E.1** (weighted graph). *Let $\hat{\mathcal{G}} = (\mathcal{V}, \mathcal{E})$ be a graph with vertices $\mathcal{V} = \mathbb{Z}_n$, edges $\mathcal{E} \subseteq \mathcal{V} \times \mathcal{V}$. Let $\hat{A} \in \mathbb{R}^{n \times n}$ denote the adjacency matrix $\hat{A}_{ij} = 1$ if $(i, j) \in \mathcal{E}$ and $0$ otherwise. We denote a weighted graph as $\mathcal{G} = (\mathcal{V}, \mathcal{E}, \mathcal{W})$ where $\mathcal{W} : \mathcal{E} \to \mathbb{R}$ are the weights of the edges. Let $A$ denote the adjacency matrix of $\mathcal{G}$, where $A_{ij} = \mathcal{W}(i, j)$ or zero if $(i, j) \notin \mathcal{E}$. The Laplacian $L = \text{Lap}(A)$ of an undirected weighted graph is defined analogous to the unweighted graph as $L = D - A$ with degree matrix elements $D_{ij} = \delta_{ik}\sum_k A_{ik}$.*

**Definition E.2** (Slow manifold). *Let $L$ be a graph Laplacian (undirected, weighted or unweighted), with spectral expansion $L = \sum_{i=1}^n \lambda_i \psi_i \psi_i^T$. Let $\varepsilon \ll 1$ and $\lambda_{\max} = \max\{\lambda_i\}$ be the largest eigenvalue of $L$. We define the slow manifold as*

$$\mathbf{Slow}_\varepsilon[L] = \text{Span}\{\psi_i | |\lambda_i| < \varepsilon^2 \lambda_{\max}\} \tag{23}$$

**Theorem E.1** (Slow modes of weighted Laplacians). *Let $A$ be the adjacency matrix of a weighted graph and $\hat{A}$ be its unweighted counterpart. Let $L = \mathrm{Lap}(A)$ and $\hat{L} = \mathrm{Lap}(\hat{A})$. Then $\mathbf{Slow}_\varepsilon[L]$ overlaps with $\mathbf{Slow}_\varepsilon[\hat{L}]$ up to $O(\varepsilon^2)$ corrections from the rest of the modes.*

To prove this we will make use of the incidence matrix representation of the Laplacian.

**Definition E.3** (Incidence matrix). *Given a weighted graph $\mathcal{G} = (\mathcal{V}, \mathcal{E}, \mathcal{W})$, define its incidence matrix as $C : \mathcal{V} \times \mathcal{E} \to \{\pm 1\}$, where for any edge $e = (i \to j) \in \mathcal{E}$, $C_{i,e} = -1$ and $C_{j,e} = 1$, and zero for other components.*

**Lemma E.2** (Laplacian in terms of the incidence matrix). *Let $w = \mathrm{vec}(\mathcal{W}(\mathcal{E}))$ be the vector of all weights indexed in the same order as the columns of $C$, with $w_e = A_{ij}$, for $e = (i, j)$ and let $W$ be a diagonal matrix with $w$ on its diagonal. Then, the Laplacian $L = \mathrm{Lap}(A)$ can be written as $L = \frac{1}{2}CWC^T$ (proof in app. F.1).*

Because $\mathcal{G}$ and $\hat{\mathcal{G}}$ share the same vertices and edges, their incidence matrix $C$ is the same. From Lemma E.2, $L = \frac{1}{2}CWC^T$ and $\hat{L} = \frac{1}{2}CC^T$ as $\hat{\mathcal{G}}$ is unweighted. Using SVD, $C = USV^T$ and defining $R = US/\sqrt{2}$ and $Q = V^TWV$, we have

$$\hat{L} = RR^T \qquad\qquad L = RQR^T. \tag{24}$$

Note that for a random configuration $X$ the edge weights $W$ will be random, as they arising from derivatives of $f_{ij}(r_{ij})$ in equation 20 (unless $f_{ij}$ is quadratic which makes $W$ constant). Therefore, we will assume $Q$ has a uniform Gaussian distribution. Assuming $W$ is also Gaussian, the spectrum of such a $Q = V^TWV$ is somewhere between the distribution of $W$ (for sparse graphs with $|\mathcal{E}| \sim O(|\mathcal{V}|)$) and a Wigner Semi-circle (for dense graphs with $|\mathcal{E}| \sim O(|\mathcal{V}|^2)$). See appendix F.2 for more discussion. We also assume $Q$ has no particular block structure and that the spectrum of any diagonal block of $Q$ should also follows a distribution similar to all of $Q$.

**Slow subspace.** We now sketch the proof for Theorem E.1. For details, refer to appendix F.4. From the SVD, $C = USV^T$, the slow subspace is

$$\mathbf{Slow}_\varepsilon[\hat{L}] = \left\{i \big| S_{ii} < \varepsilon \max[S]\right\} \tag{25}$$

Normalize $\hat{S} = S/\max[S]$ and make them all positive (e.g. absorb their sign into $U$). For some $\varepsilon < 1$ sort the SV such that $\hat{S} = \mathrm{diag}(S_\varepsilon, S_1)$ where the diagonal matrices $S_\varepsilon < \varepsilon$ and $S_1 \geq \varepsilon$. Now, the problem of finding $\mathbf{Slow}_\varepsilon[L]$ becomes finding eigenvectors of the matrix $\hat{M} = \hat{S}Q\hat{S}^T$ with eigenvalues $O(\varepsilon^2)$. Using $S_\varepsilon \sim O(\varepsilon)$ and $S_1 \sim O(1)$, we can pull factors of $\varepsilon$ out from $\hat{M}$ and write it as

$$\hat{M} = M_0 + \hat{\varepsilon}\delta M, \qquad M_0 = \begin{pmatrix} \hat{\varepsilon}^2\hat{A} & 0 \\ 0 & C \end{pmatrix}, \qquad \delta M = \begin{pmatrix} 0 & \hat{B} \\ \hat{B}^T & 0 \end{pmatrix}. \tag{26}$$

where $\hat{\varepsilon}^2 \equiv \varepsilon^2\sqrt{n_A/n_C}$ is rescaled so that the random matrices $\hat{A} \in \mathbb{R}^{n_A \times n_A}$ and $C \in \mathbb{R}^{n_C \times n_C}$ have a similar range of eigenvalues. Next, using a perturbative ansatz for eigenvectors $\psi' = \psi + \hat{\varepsilon}\delta\psi$ and eigenvalues $\lambda' = \lambda + \hat{\varepsilon}\delta\lambda$, we solve $\hat{M}\psi' = \lambda'\psi'$ up to $O(\hat{\varepsilon}^2)$ corrections.

To find slow modes for $L$ we start from $\psi \in \mathbf{Slow}_\varepsilon[\hat{L}]$. Specifically, we start with an eigenvector $\psi_A$ of $\hat{A}$ and concatenate it with zeros to get $\psi = (\psi_A, 0)$. We have $M_0\psi = \lambda\psi$ with $\lambda = \hat{\varepsilon}^2\lambda_A$. Using first-order perturbation theory, we find the corrections $\delta\lambda$ to the eigenvalues and eigenvectors to be

$$\delta\lambda = \psi^T\delta M\psi = 0, \qquad \delta\psi = -(M_0 - \lambda)^{-1}\delta M\psi = \begin{pmatrix} 0 \\ (C - \lambda)^{-1}\hat{B}^T\psi_A \end{pmatrix}. \tag{27}$$

Putting all together we find the slow eigenvector $\psi' = \psi + \hat{\varepsilon}\delta\psi$ up to order $O(\varepsilon^2)$ to be

$$\mathbf{Slow}_\varepsilon[L] \ni \psi' = \begin{pmatrix} \psi_A \\ \hat{\varepsilon}(C - \hat{\varepsilon}^2\lambda_A)^{-1}\hat{B}^T\psi_A \end{pmatrix}, \qquad \hat{M}\psi' = \hat{\varepsilon}^2\lambda\psi' + O(\hat{\varepsilon}^2) = O(\hat{\varepsilon}^2) \tag{28}$$

meaning to first order in $\hat{\varepsilon}$ the corrections to eigenvalues of slow modes vanishes. This is desired because the slow mode eigenvalues are $O(\hat{\varepsilon}^2)$. We also observe that slow modes of $L$ are mostly confined to $\mathbf{Slow}_\varepsilon[\hat{L}]$ and only get $O(\varepsilon)$ contributions from the fast subspace of $\hat{L}$.

As a side, it follows that all weighted graphs share the null space of the unweighted Laplacian.

**Proposition E.3** (Shared null space). *Let* $\mathbf{Null}[M] = \mathrm{Span}\{v|v \in \mathbb{R}^n, Mv = 0\}$ *denote the null space of a matrix* $M \in \mathbb{R}^{n \times n}$. *The null space of the Laplacian* $\hat{L}$ *(unweighted) is contained in the null space of Laplacian* $L$ *(weighted), meaning* $\mathbf{Null}[\hat{L}] \subseteq \mathbf{Null}[L]$.

**Lemma E.4.** $\mathbf{Null}[\hat{L}] = \mathbf{Null}[R^T]$

*Proof.* $\forall v \in \mathbf{Null}[\hat{L}], 0 = v^T \hat{L} v = \|\mathbb{R}^T v\|^2$ and $\forall v \in \mathbf{Null}[R^T]$, $\hat{L}v = RR^T v = 0$. □

*Proof of proposition E.3.* $\forall v \in \mathbf{Null}[\hat{L}]$, $Lv = RQR^T v = 0$ hence, $\mathbf{Null}[\hat{L}] \subseteq \mathbf{Null}[L]$. □

Note that $\mathbf{Null}[\hat{L}]$ and $\subseteq \mathbf{Null}[L]$ are not necessarily the same because weights can be zero, which could make the null space of the weighted graph larger than the unweighted one. Next, we present our method for coarse-graining using a set of canonical slow modes.

## F   INVARIANT ADDITIVE DYADIC POTENTIALS

We want to Compute the Hessian of equation 17, $\mathscr{L}(X) = \sum_{ij}(r_{ij})$. Let $\hat{r} = \eta \boldsymbol{r}/r$ be the dual unit vector of $\boldsymbol{r}$. First, note that

$$\partial_i r_{jk} \equiv \frac{\partial r_{jk}}{\partial X_i} = \partial_i \sqrt{\|X_j - X_k\|_\eta}$$
$$= \eta \frac{\boldsymbol{r}_{jk}}{r_{jk}}(\delta_{ij} - \delta_{ik}) = \hat{r}_{jk}(\delta_{ij} - \delta_{ik}) \qquad (29)$$

Then the gradient becomes

$$\partial_i \mathscr{L}(X) = \sum_{j,k} f'_{jk}(r_{jk}) \frac{\partial r_{jk}}{\partial x_i}$$
$$= \sum_{j,k} f'_{jk}(r_{jk}) \eta \hat{r}_{jk}(\delta_{ij} - \delta_{ik})$$
$$= 2 \sum_j f'_{ij}(r_{ij}) \eta \hat{r}_{ij}. \qquad (30)$$

where we used $\hat{r}_{jk} = -\hat{r}_{kj}$ to show both terms in $(\delta_{ij} - \delta_{ik})$ yield the same output. Finally, the Hessian becomes

$$[H(X)]_{ij} = \partial_i \partial_j \mathscr{L}(X) = 2 \partial_j \sum_k f'_{ik}(r_{ik}) \hat{r}_{ik}$$

$$= 2 \sum_k [f''_{ik}(r_{ik}) \partial_j r_{ik} \otimes \hat{r}_{ik} + f'_{ik}(r_{ik}) \partial_j \hat{r}_{ik}]$$

$$= 2 \sum_k \Big[ (\delta_{ji} - \delta_{jk}) f''_{ik}(r_{ik}) \hat{r}_{ik} \otimes \hat{r}_{ik}$$
$$+ f'_{ik}(r_{ik}) \Big( \eta \frac{\delta_{ji} - \delta_{jk}}{r_{ik}} - \frac{\hat{r}_{ik}}{r_{ik}^2} \partial_j r_{ik} \Big) \Big]$$

$$= 2 \sum_k (\delta_{ji} - \delta_{jk}) \Big[ f''_{ik}(r_{ik}) \hat{r}_{ik} \otimes \hat{r}_{ik} + f'_{ik}(r_{ik}) \Big( \frac{\eta}{r_{ik}} - \frac{\hat{r}_{ik}}{r_{ik}^2} \otimes \hat{r}_{ik} \Big) \Big]$$

$$= 2 \sum_k \Big[ f''_{ik}(v) \hat{v} \otimes \hat{v} + \frac{f'_{ik}(v)}{v} (\eta - \hat{v} \otimes \hat{v}) \Big]_{\boldsymbol{v} = \boldsymbol{r}_{ik}} (\delta_{ij} - \delta_{jk})$$

$$= 2 \sum_k \Big[ \Big( f''_{ik}(v) - \frac{f'_{ik}(v)}{v} \Big) \hat{v} \otimes \hat{v} + \frac{f'_{ik}(v)}{v} \eta \Big]_{\boldsymbol{v} = \boldsymbol{r}_{ik}} (\delta_{ij} - \delta_{jk})$$

$$= \sum_k \boldsymbol{H}_{ik}(x) (\delta_{ij} - \delta_{jk}) = \mathrm{Lap}(\boldsymbol{H})_{ij} \qquad (31)$$

This is because the components of Laplacian can be be written

$$
\begin{aligned}
L_{ij} = \mathrm{Lap}(A)_{ij} &= (D - A)_{ij} \\
&= \delta_{ij} \sum_k A_{ik} - A_{ij} = \sum_k A_{ik}(\delta_{ij} - \delta_{jk})
\end{aligned}
\tag{32}
$$

### F.1    INCIDENCE MATRIX

The Laplacian $L = D - A$ of an undirected graph with adjacency $A$ can be written as $L = CWC^T/2$ using the incidence matrix $C$ and the edge weights $W$. This can be shown as follows

$$
\begin{aligned}
[CWC^T]_{ij} &= \sum_e C_i^e W_{ee} C_j^e \\
&= \sum_{k,l} C_i^{(k \to l)} A_{kl} C_j^{(k \to l)} \\
&= \sum_{k,l} (\delta_{il} - \delta_{ik}) A_{kl} (\delta_{jl} - \delta_{jk}) \\
&= \sum_{k,l} (\delta_{il}\delta_{jl} - \delta_{ik}\delta_{jl} - \delta_{il}\delta_{jk} + \delta_{ik}\delta_{jk}) A_{kl} \\
&= 2 \sum_{k,l} (\delta_{il}\delta_{jl} - \delta_{ik}\delta_{jl}) A_{kl} \\
&= 2 \sum_k \delta_{ij} A_{kj} - 2 A_{ij} = 2(D - A)_{ij} = 2 L_{ij}
\end{aligned}
\tag{33}
$$

where we assumed $A_{kl} = A_{lk}$ (undirected graph).

So the same derivation of the backbone also holds for this case. The idea is that using the incidence matrix $C$ and edge weights $W$ (as a diagonal matrix), any Laplacian $L$ can be decomposed as $L = CWC^T$. Then, doing SVD $C = USV^T$ we have

$$
L = USV^T WVS^T U^T = UMU^T
\tag{34}
$$

Where the matrix $M = SV^T WVS^T$ has an interesting property, namely that its null space includes the null space of the unweighted Laplacian $L_0 = CC^T$. To see this note that $L_0 = USS^T S^T$, which means columns $U_i$ are the eigenvectors of $L_0$ with eigenvalues $S_i^2$. The null eigenspace of $L_0$ are the $U_i$ for which $S_i = 0$. This subspace will also be a null subspace for $L$, because that block is also zero in $M$, because $M_{ij} = \sum_c S_i V_{ik} W_{kk} V_{jk} S_j$. So, whenever $S_i = 0$ or $S_j = 0$, $M_{ij} = 0$, meaning that whole block in $M$ is zero and $MU_i = 0$ ( write it better).

**Example: power law.**    Let $f(r) = r^p$. We have $f' = pr^{p-1}$ and $f'' = p(p-1)r^{p-2}$, yielding the Hessian

$$
\boldsymbol{H} = \nabla\nabla f(r) = r^{p-2} \left[ \left( p^2 - 2p \right) \hat{r} \otimes \hat{r} + p\eta \right]
\tag{35}
$$

$$
B_{ik} = A_{ik} r_{ik}^{p-2} \left[ \left( p^2 - 2p \right) \hat{r}_{ik} \otimes \hat{r}_{ik} + p\eta \right]
\tag{36}
$$

**Example: Lennard-Jones.**    This potential has the form

$$
f(r) = 4\varepsilon \left[ \left( \frac{\sigma}{r} \right)^p - \left( \frac{\sigma}{r} \right)^q \right]
\tag{37}
$$

where for classic van-der Waals potential $p = 2q = 12$. The Hessian for this potential is given by

$$
\boldsymbol{H}(r) = \nabla\nabla f(r) = \varepsilon \left[ \left( \frac{\sigma}{r} \right)^{p+2} \left[ \left( p^2 + 2p \right) \hat{r} \otimes \hat{r} - p\eta \right] - \left( \frac{\sigma}{r} \right)^{q+2} \left[ \left( q^2 + 2q \right) \hat{r} \otimes \hat{r} - q\eta \right] \right]
\tag{38}
$$

and $B_{ik} = A_{ik} \boldsymbol{H}(r_{ik})$

### F.2 STRUCTURE AND SPECTRUM OF OF $Q = V^T W V$

To consider only the relevant subspace of SVD, we have $U, S \in \mathbb{R}^{n \times n}$, and $V \in \mathbb{R}^{m \times n}$, with $n = |\mathcal{V}|$ and $m = |\mathcal{E}|$. For a connected undirected graph $m \geq 2(n - 1)$ and $V$ is full rank ($V^T V = I_n$). Note the edge weights $W$ come from the forces $f_{ij}(r_{ij})$ in equation 20, which for an arbitrary $X$ will be random. Assuming a Gaussian distribution $W_{ee} \sim \mathcal{N}(0, \sigma)$ for all edges $e$, the matrix $Q$ will also have random Gaussian entries. When $m = n$, $V$ defines the eigenbasis of $Q$ and $W_{ee}$ are the eigenvalues of $Q$. Similarly, in sparse graphs, where $m \sim O(n)$, $V$ is approximately the eigenbasis and the spectrum of $Q$ should have a distribution similar to $W_{ee}$. For dense graphs, where $m \sim O(n^2)$, every entry of $Q$ will involve a weighted sum over multiple $W_{ee}$. Then, from central limit theorem, entries of $Q$ will asymptotically have a Gaussian distribution. From random matrix theory, we know that such $Q$ will have a spectrum which follows the Wigner-semi-circle law. In both cases (sparse and dense graphs) the spectrum of $Q$ has a finite variance and sits somewhere between a Gaussian and a semi-circle.

### F.3 GENERALIZATION TO NONZERO BUT SMALL SV

We want to know how much the slow modes of weighted and unweighted graphs to overlaps. With the spectral expansion $\hat{L} = \sum_i \lambda_i \psi_i \psi_i^T$ Define the slow subspace as in equation 23

$$\mathbf{Slow}_\varepsilon[\hat{L}] = \mathrm{Span}\{\psi_i | |\lambda_i| < \varepsilon^2 \lambda_{\max}(\hat{L})\} \tag{39}$$

where $\lambda_{\max}(\hat{L}) = \max\{\lambda_i\} = \max_\psi[\psi^T L \psi / \|\psi\|^2]$ is the largest eigenvalue of $L$ and $\varepsilon \ll 1$. In terms of the singular values (SV) of the incidence matrix $C = USV^T$, the slow subspace becomes

$$\mathbf{Slow}_\varepsilon[\hat{L}] = \{i | S_{ii} < \varepsilon \max[S]\} \tag{40}$$

We will show that the slow modes in weighted $L = CWC^T$ are perturbations to the slow modes of $\hat{L}$. Define

$$M = SV^T W V S^T = SQS^T \tag{41}$$

Normalize $\hat{S} = S/\max[S]$. Break the space down to the slow and fast subspaces, based on whether $\hat{S}_{ii} < \varepsilon$ or not. First, since $L$ is positive semi-definite, we can make all $S_{ii} \geq 0$. Let $\hat{S} = S/\max S$. We sort the dimensions in $\hat{S}$ to have the small SVs appear first. Denote the block in $\hat{S}$ where $S_{ii} < \varepsilon$ by $S_\varepsilon$. We have

$$\hat{S}^2 = \begin{pmatrix} S_\varepsilon^2 & 0 \\ 0 & S_1^2 \end{pmatrix} < \begin{pmatrix} \varepsilon^2 & 0 \\ 0 & 1 \end{pmatrix} \tag{42}$$

We know the null space of $\hat{L}$, where $S_{ii} = 0$, is shared with $L$. First, we remove the null space from $L$ and $\hat{L}$, calling the remainder $L_0$ and $\hat{L}_0$ and the remaining SVs $\hat{S}$. Then in this remainder subspace we need to find parts which are $O(\varepsilon)$. We sort the dimensions in $\hat{S}$ to have the small SVs appear first. We denote the block in $\hat{S}$ where $S_{ii}^2 < \varepsilon \max[S^2]$ by $S_\varepsilon$. We have

$$M = \max[S]^2 \hat{S} Q \hat{S}^T = \begin{pmatrix} S_\varepsilon Q_{\varepsilon\varepsilon} S_\varepsilon & S_\varepsilon Q_{\varepsilon 1} S_1 \\ S_1 Q_{\varepsilon 1}^T S_\varepsilon & S_1 Q_{11} S_1 \end{pmatrix} = \begin{pmatrix} M_{\varepsilon\varepsilon} & M_{\varepsilon 1} \\ M_{\varepsilon 1}^T & M_{11} \end{pmatrix} \tag{43}$$

Because $S_\varepsilon$ is $O(\varepsilon)$ and $S_1$ is $O(1)$, we will factor out the factors of $\varepsilon$ from blocks in $M$ and write

$$M = \max[S]^2 \begin{pmatrix} \varepsilon^2 A & \varepsilon B \\ \varepsilon B^T & C \end{pmatrix} \tag{44}$$

Here $A$ and $C$ are random matrices built from their corresponding blocks in $Q$ and sandwiched between $S_\varepsilon/\varepsilon$ (for $A$), and $S_1$ (for $C$), which have $O(1)$ values. The spectrum of $Q$ has a distribution between a Gaussian with mean zero and a Wigner semi-circle, also centered around zero. We expect spectra of $A$ and $C$ to be similar to $Q$. Denote the spectral expansion of $Q$ as

$$Q = \Psi \Lambda \Psi^T, \qquad \Lambda = \mathrm{diag}(\lambda_i)_{i=1}^n, \qquad \Psi = [\psi_i]_{i=1}^n. \tag{45}$$

This is because when $Q_{ij} \sim \mathcal{N}(0, \sigma)$ we have (ignoring Bessel's correction for $k \gg 1$).

$$\sigma^2 = \text{Var}(Q_{ij}) \approx \frac{1}{n}\|Q\|^2 = \frac{1}{n}\sum_i \lambda_i^2 = \text{Var}(\Lambda) \tag{46}$$

where we assumed $\text{Tr}\{Q\}/n \approx \text{mean}(Q) = 0$. Since a block $Q_k$ of size $k$ is $k^2$ entries sampled from the same distribution as $Q$, we expect

$$\frac{\|Q_k\|^2}{k^2} \approx \frac{\|Q_l\|^2}{l^2} \quad \Rightarrow \frac{1}{k}\text{Var}(Q_k) \approx \frac{1}{l}\text{Var}(Q_l) \tag{47}$$

Thus, rescaling $A \in \mathbb{R}^{n_A \times n_A}$ and $C \in \mathbb{R}^{n_C \times n_C}$ we get

$$\hat{A} = \frac{A}{\sqrt{n_A}}, \quad \hat{C} = \frac{C}{\sqrt{n_C}}, \quad \text{Var}(\hat{A}) \approx \text{Var}(\hat{C}) \tag{48}$$

### F.4    Approximate slow modes of $L$

If $M$ did not have the off-diagonal blocks $B$, then $\textbf{Slow}_\varepsilon[L]$ and $\textbf{Slow}_\varepsilon[\hat{L}]$ would coincide, as the $S_\varepsilon$ block and the $S_1$ block would not mix when $B = 0$. Define $M_0$ as the block matrix of $M$ with $B = 0$.

$$M_0 \equiv \begin{pmatrix} \varepsilon^2 A & 0 \\ 0 & C \end{pmatrix} \tag{49}$$

Using spectral expansions

$$A = \Psi_A \Lambda_A \Psi_A^T, \quad C = \Psi_C \Lambda_C \Psi_C^T \tag{50}$$

the eigenvectors of $M_0$ consist of

$$M_0 \begin{pmatrix} \psi_{Ai} \\ 0 \end{pmatrix} = \varepsilon^2 \lambda_{Ai} \begin{pmatrix} \psi_{Ai} \\ 0 \end{pmatrix}, \qquad M_0 \begin{pmatrix} 0 \\ \psi_{Ci} \end{pmatrix} = \lambda_{Ci} \begin{pmatrix} 0 \\ \psi_{Ci} \end{pmatrix}. \tag{51}$$

Since we are looking for slow modes, we must also consider the magnitudes of $\lambda_{Ai}$ and $\lambda_{Ci}$. Since $A$ and $C$ entries are random samples from $Q$, we expect them to have a semi-circle or Gaussian distribution similar to $Q$. Thus, we can use the variances of eigenvalues of $A$ and $C$ as a proxy for the how the magnitudes of $\lambda_{Ai}$ and $\lambda_{Ci}$ compare. From equation 48 we have

$$\frac{1}{n_A}\mathbb{E}[\Lambda_A^2] \approx \frac{1}{n_A}\text{Var}(A) \approx \frac{1}{n_C}\mathbb{E}[\Lambda_C^2] \tag{52}$$

Based on this we define a rescaled $\hat{\varepsilon}$ such that $\varepsilon^2 \lambda_{Ai}$ still has a smaller magnitude than $\lambda_{Ci}$ on average, meaning we want

$$\varepsilon^4 \mathbb{E}[\Lambda_A^2] < \mathbb{E}[\Lambda_C^2] \qquad \Rightarrow \qquad \varepsilon^4 n_A < n_C \qquad \Rightarrow \qquad \hat{\varepsilon}^2 \equiv \varepsilon^2 \sqrt{\frac{n_A}{n_C}} < 1 \tag{53}$$

We choose $\varepsilon$ such that the condition in equation 53 is satisfied. We can express $M$ in terms of $\hat{\varepsilon}$ by rescaling $A$ and $B$ to $\hat{\varepsilon}^2 \hat{A} = \varepsilon^2 A$ and $\hat{\varepsilon}\hat{B} = \varepsilon B$. Now eigenvalues of $\hat{A}$ have the same variance as eigenvalues of $C$. For brevity, denote $\hat{M} = \max[S]^2 M$. We have

$$\hat{M} = \begin{pmatrix} \hat{\varepsilon}^2 \hat{A} & \hat{\varepsilon}\hat{B} \\ \hat{\varepsilon}\hat{B}^T & C \end{pmatrix}. \tag{54}$$

To find how slow modes of $\hat{M} = SQS^T/\max[S]^2$ differ from slow modes of $SS^T$, we break $\hat{M}$ into a block diagonal part and an $O(\hat{\varepsilon})$ off-diagonal perturbation

$$\hat{M} = M_0 + \hat{\varepsilon}\delta M, \qquad M_0 = \begin{pmatrix} \hat{\varepsilon}^2 \hat{A} & 0 \\ 0 & C \end{pmatrix} \qquad \delta M = \begin{pmatrix} 0 & \hat{B} \\ \hat{B}^T & 0 \end{pmatrix}. \tag{55}$$

As in equation 51, eigenvectors of $A = \sqrt{n_C/n_A}\hat{A}$ and $C$ are eigenvectors of $M_0$. Now we want to find eigenvectors of $\hat{M}$ with small $O(\varepsilon^2)$ eigenvalues up to order $\hat{\varepsilon}$ corrections by treating $\delta M$ as a perturbation.

$$(M_0 + \hat{\varepsilon}\delta M)(\psi + \hat{\varepsilon}\delta\psi) = (\lambda + \hat{\varepsilon}\delta\lambda)(\psi + \hat{\varepsilon}\delta\psi)$$
$$M_0\psi + \hat{\varepsilon}(\delta M\psi + M_0\delta\psi) + O(\hat{\varepsilon}^2) = \lambda\psi + \hat{\varepsilon}(\delta\lambda\psi + \lambda\delta\psi) + O(\hat{\varepsilon}^2)$$
$$\Rightarrow \delta M\psi + M_0\delta\psi = \delta\lambda\psi + \lambda\delta\psi \tag{56}$$

We only need the components of $\delta\psi$ orthogonal to $\psi$, so we can assume $\delta\psi^T\psi = 0$. From this we have

$$\delta\lambda = \psi^T\delta M\psi + \psi^T M_0\delta\psi = \psi^T\delta M\psi, \tag{57}$$

where we used $\psi^T M_0\delta\psi = \lambda\psi^T\delta\psi = 0$. Plugging equation 57 into equation 56 we can solve for $\delta\psi$ by inverting the matrices

$$(M_0 - \lambda)\delta\psi = (\delta\lambda - \delta M)\psi$$
$$\Rightarrow \delta\psi = (M_0 - \lambda + i\eta)^{-1}(\delta\lambda - \delta M)\psi \tag{58}$$

where we added a small $\eta$ to make the matrix $M_0 - \lambda$ invertible, as $\lambda$ is one of its eigenvalues.

To find slow modes, we start from slow modes of $M_0$ which are in the $A$ subspace. Let $\psi_A$ be an eigenvector of $A$ with $\hat{A}\psi_A = \lambda_A\psi_A$. Concatenating $\psi_A$ with zeros in the $C$ subspace we have

$$\psi = \begin{pmatrix}\psi_A \\ 0\end{pmatrix}, \qquad\qquad M_0\psi = \hat{\varepsilon}^2\lambda_A\psi. \tag{59}$$

Using this $\psi$ to compute $\delta\lambda$ in equation 57 we have

$$\delta\lambda = \psi^T\delta M\psi = \begin{pmatrix}\psi_A^T & 0\end{pmatrix}\begin{pmatrix}0 \\ \hat{B}^T\psi_A\end{pmatrix} = 0 \tag{60}$$

meaning to first order in $\hat{\varepsilon}$ the corrections to eigenvalues of slow modes vanishes. This is desired because the slow mode eigenvalues are $O(\hat{\varepsilon}^2)$ and we find that with this $\psi$ ansatz the corrections it will get are also at least $O(\hat{\varepsilon}^2)$. Next, we compute the corrections $\delta\psi$ to the eigenvectors. Plugging $\psi$ into equation 58 with $\lambda = \hat{\varepsilon}^2\lambda_A$ and $\delta\lambda = 0$ we have

$$(M_0 - \lambda + i\eta)^{-1} = \begin{pmatrix}(\hat{\varepsilon}^2\hat{A} - \lambda + i\eta)^{-1} & 0 \\ 0 & (C - \lambda)^{-1}\end{pmatrix}$$
$$\delta\psi = -(M_0 - \lambda + i\eta)^{-1}\delta M\psi$$
$$= \begin{pmatrix}0 \\ (C - \lambda)^{-1}\hat{B}^T\psi_A\end{pmatrix} \tag{61}$$

where we dropped $i\eta$ in the lower block because $\hat{\varepsilon}^2\lambda_A$ is unlikely to be also an eigenvalue of $C$, as $A$ and $C$ are random matrices.

Using the relation $\hat{\varepsilon}\hat{B} = \varepsilon B$ with the original $\varepsilon$ and putting all together we find the eigenvector $\psi' = \psi + \hat{\varepsilon}\delta\psi$ up to order $O(\varepsilon^2)$ to be

$$\psi' = \begin{pmatrix}\psi_A \\ \hat{\varepsilon}(C - \hat{\varepsilon}^2\lambda_A)^{-1}\hat{B}^T\end{pmatrix} \tag{62}$$
$$\hat{M}\psi' = \hat{\varepsilon}^2\lambda\psi' + O(\hat{\varepsilon}^2) = O(\hat{\varepsilon}^2) \tag{63}$$

