# OpenReview forum: "Hessian Reparametrization for Coarse-grained Energy Minimization"
_ICLR.cc/2024/Workshop/AI4DiffEqtnsInSci — AI4DiffEqtnsInSci @ ICLR 2024 Poster_

### Official Review · Reviewer_6xz4 · 2024-02-15
**I find it very difficult to follow the logic of the current manuscript**

**Rating:** 3
**Confidence:** 4

**Review:**

The authors introduced a novel method that parameterizes a high-dimensional parameter of interest ($X$) via a low-dimensional variable ($Z$). Specifically, they propose to use slow-mode projection and GNN for this reparameterization. Then they introduce the Hessian backbone. My comments are below:

- The current manuscript does not flow well. A lot of sections seem disjunct. For example, it should be mentioned in section 1.1 how slow mode projection method compares to GNN. One is linear and the other is nonlinear? Which one should I choose? Why are two of these methods listed here all of a sudden? Then the authors introduced Hessian backbone in section 1.2. I do not see any connection between section 1.1 and 1.2 and I sincerely could not follow from the Hessian section to the end.
- I found the idea of $X=GNN(Z)$ very similar to deep image priors (https://arxiv.org/abs/1711.10925) and their applications on high-dimensional Bayesian inference (e.g. MCMC (https://library.seg.org/doi/10.1190/geo2021-0666.1) or gradient-based optimization (https://agupubs.onlinelibrary.wiley.com/doi/full/10.1029/2022JB025964)). I suggest authors briefly mention these prior arts in their introduction, and see how their methods compare to those.
- On that note, is the dimension of GNN weights smaller than the dimension of $X$?
- I also found the slow mode projection method introduced in section 1.1 very close to model order reduction and proper orthogonal decomposition.
- In traditional approaches, is $\mathcal{L}$ usually assumed to be expensive while $\mathcal{L}_{\mathrm{CG}}$ is not? Suggest clarification.
- However, for the CG methods introduced in this manuscript (either slow mode projection or GNN), the cost of the coarse-scale and fine-scale simulators (loss functions) are almost the same --- because $\mathcal{L}_{\mathrm{CG}}=\mathcal{L} \circ \rho$. Then, it is hard to justify the benefit of using CG one.

---

### Official Review · Reviewer_o5Jn · 2024-02-26
**Review comments**

**Rating:** 7
**Confidence:** 2

**Review:**

The reviewer, while not an expert in statistical physics, offers comments from a machine learning standpoint. The paper explores a novel coarse-graining method via reparametrization, circumventing the 'Force-matching' (dimensional reduction) and 'Back-mapping' (reconstruction) steps by employing a reparametrization X = ρ(Z), where ρ : Z → X. However, the reviewer perceives this process as analogous to dimensional reduction in machine learning and akin to the 'Force-matching' concept, thus finding limited novelty from a learning perspective.

Nevertheless, the discussion on physical Hessians proves to be intriguing and thought-provoking. The author demonstrates that many physical problems exhibit a distinctive structure within the Hessian matrix, leveraging this property to devise an algorithm that projects onto the slow manifold.

---

### Meta-Review · Area_Chair_fM7e · 2024-02-28

**Recommendation:** Accept (Poster)

**Metareview:**

Dear Authors,

Thank you for submitting the draft.

Both reviewers agree that the presented work presents some interesting strengths. However, both reviewers do also raise some major points of concern, especially regarding the clarity of the presentation and some of the claims made. It is expected that authors will be addressing comments by the reviewers in the final draft.

regards

AC

---

### Decision · Program_Chairs · 2024-02-29

Accept (Poster)